# Immune Response Against Influenza in a Cohort of Repeatedly Vaccinated Adults During the 2017/2018 and 2018/2019 Seasons

**DOI:** 10.3390/vaccines12111218

**Published:** 2024-10-26

**Authors:** Raquel Guiomar, Susana Pereira da Silva, Ana Paula Rodrigues, Inês Costa, Patrícia Conde, Paula Cristóvão, Pedro Pechirra, Paulo Estragadinho, Kamal Mansinho, Olav Hungnes, António Silva Graça, Baltazar Nunes

**Affiliations:** 1National Reference Laboratory for Influenza and Other Respiratory Viruses, Infectious Diseases Department, National Institute of Health Dr. Ricardo Jorge, 1649-016 Lisbon, Portugal; ines.costa@insa.min-saude.pt (I.C.); patricia.conde@iniav.pt (P.C.); paula.cristovao@iniav.pt (P.C.); pedro.pechirra@ipma.pt (P.P.); 2National School of Public Health, NOVA University, 1600-560 Lisbon, Portugal; bnunes@ensp.unl.pt; 3Department of Epidemiology, National Institute of Health Dr. Ricardo Jorge, 1649-016 Lisbon, Portugal; susana.pereira@insa.min-saude.pt (S.P.d.S.); ana.rodrigues@insa.min-saude.pt (A.P.R.); 4Department of Occupational Medicine, National Institute of Health Dr. Ricardo Jorge, 1649-016 Lisbon, Portugal; paulo.estragadinho@insa.min-saude.pt (P.E.); silva.graca@insa.min-saude.pt (A.S.G.); 5Centro Hospitalar de Lisboa Ocidental, 1449-005 Lisbon, Portugal; 6Norwegian National Influenza Centre, Norwegian Institute of Public Health, 0213 Oslo, Norway; olav.hungnes@fhi.no

**Keywords:** influenza vaccine, antibodies, cohort study, vaccination history, frequently and occasionally vaccinated

## Abstract

Background/Objectives: The influenza vaccination of healthcare workers (HCWs) is recommended each autumn and winter season by the relevant authorities in EU/EEA countries. The objective of this study was to evaluate the impact of repeated trivalent influenza vaccine (TIV) uptake during the 2017/2018 and 2018/2019 seasons on vaccine-derived immunity against influenza. Methods: A cohort study of HCWs vaccinated with an annual TIV was conducted from October 2017 to June 2019. The protective antibodies against the influenza vaccine strains were assessed at three time points: prior to vaccination and at one and six months following vaccination for each season. Sera were tested by hemagglutination inhibition assay. Participants were grouped according to their history of TIV vaccination over four seasons (since 2015/16), with the groups designated as “frequently vaccinated” (≥3 vaccines) and “occasionally vaccinated” (≤2 vaccines). Seroprevalence, geometric mean titer (GMT) and seroconversion rate were compared between the frequently and occasionally vaccinated groups. Results: A total of 97 healthcare workers (HCWs) were enrolled in the study; 49 HCWs participated in both seasons. Thirty-two (43.2%) and forty-three (59.7%) individuals had ≥3 vaccines since 2015/2016, at recruitment and during the 2017/2018 and 2018/2019 influenza seasons, respectively. One month following vaccination, HCWs who had received occasional vaccinations demonstrated a higher prevalence of protective antibodies and a greater GMT for both influenza A(H1N1)pdm09 and A(H3N2) viruses. For influenza B Victoria, the frequently vaccinated HCWs demonstrated a higher seroprevalence rate, seroconversion, and GMT. Conclusions: Previous vaccination can influence the immune response, although without substantially compromising the immunogenicity of annual influenza vaccination. HCW annual influenza vaccination is required to re-establish and maintain the antibody titers against influenza.

## 1. Introduction

Vaccines are one of the most effective methods for disease prevention, reducing the risk and complications associated with influenza virus infection [1]. Due to the nature of the influenza virus and its ability to evolve through the acquisition of mutations, the vaccine is annually updated to maintain protection against viruses in circulation [2]. In Portugal, the vaccine is recommended for individuals over the age of six months who have immunosuppression or chronic diseases, for the general population over the age of 60 years old, and also for healthcare workers (HCWs), in accordance with the national guidelines [3]. In Portugal, between 2017 and 2019, the trivalent vaccine against influenza (TIV), comprising strains endorsed by the World Health Organization (WHO) for the influenza A viruses A(H3N2), A(H1N1)pdm09, and B/Victoria lineage, was recommended [4]. Following the administration of the influenza vaccine, it is anticipated that an immune response will be mounted against the various types and subtypes of the influenza virus. It is widely recognized that the immunological response may vary according to age and the history of previous infections and vaccination [5]. Since 2017, the WHO stated that serological studies are of great importance, contributing to a better knowledge of the host determinants for the response to vaccines and to infection. This may support public health decisions for the prevention and control of influenza [6]. The National Institute of Health Doctor Ricardo Jorge (INSA), within the scope of its mission as a reference laboratory coordinating the National Influenza Surveillance Program for Influenza and Other Respiratory Viruses [7], implemented a study to evaluate the immune response and waning antibody titers in HCWs who received the influenza seasonal vaccine during two consecutive seasons, 2017/2018 and 2018/2019.

The present study assessed the impact of influenza seasonal vaccination on acquired immunity and monitored antibody dynamics over time. The aim was to investigate whether repeated TIV vaccination impairs vaccine-derived immunity against influenza. Furthermore, the impact of the 2009 monovalent pandemic vaccine (2009PV) uptake on immunogenicity against the TIV A(H1N1)pdm09 virus was evaluated eight years later.

## 2. Materials and Methods

### 2.1. Study Design

From October 2017 to June 2019 (2017/2018 and 2028/2019 influenza seasons), we conducted a cohort study of vaccinated HCWs at INSA. We aimed to evaluate the immunogenicity of TIV one and six months after vaccine administration among HCWs according to s previous vaccination history since 2015/2016 and the long-term effect of 2009PV vaccination on A(H1N1)pdm09 TIV immune response.

In each season, 120 HCWs were vaccinated and invited to participate in the study. A convenience sample was employed, with all HCWs who consented to participate being included. Written informed consent was provided by all HCWs during the enrollment at the Department of Occupational Medicine. The study was approved by the Ethical Commission for Health of INSA, approval references CES.35.2017 and CES.68.2018.

All HCWs received at least one dose of the TIV (influenza vaccine containing inactivated surface antigen from 3 viruses, 15 micrograms hemagglutinin) in October–November 2017 and/or in November–December 2018. For each participant, a venous blood sample was collected at day 0 (pre-vaccination), 1 month, and 6 months following vaccination in each season. During the follow-up period, a nasopharyngeal swab was collected whenever participants self-reported influenza-like illness (ILI) symptoms [8]. Laboratory diagnosis was performed for influenza using biomolecular methods (RT-PCR) [2]. Influenza-positive cases were excluded from the next follow-up points in each season. We collected data on participants’ age, sex, comorbidities, and influenza vaccine history (TIV since 2015/2016 and the 2009PV uptake). Vaccination history was self-reported at recruitment and confirmed by medical records at the Department of Occupational Medicine.

At recruitment, participants were classified according to previous TIV vaccinations from 2015/2016 as frequently vaccinated (≥3 vaccines) or occasionally vaccinated (≤2 vaccines) (Appendix A).

To evaluate the impact of 2009PV vaccination on the immunogenicity against the TIV A(H1N1)pdm09 virus eight years after vaccination, we assessed the immunological response to A(H1N1)pdm09 TIV virus (A/Michigan/45/2015) and A(H1N1)pdm09 2009PV virus (A/California/7/2009).

### 2.2. Immune Response Against Influenza Vaccine Strains

Antibody responses to influenza viruses recommended for the influenza vaccine composition for the northern hemisphere, 2017/2018 and 2018/2019 influenza seasons—A/Michigan/45/2015(H1N1)pdm09, A/Hong Kong/4801/2014(H3N2), B/Brisbane/60/2008 (Victoria) and A/Michigan/45/2015(H1N1)pdm09, A/Singapore/INFIMH-16-0019/2016(H3N2), and B/Colorado/06/2017(Victoria)—were measured by hemagglutination inhibition (HI) assays (Appendix A). Additionally, antibody response against A/California/7/2009(H1N1)pdm09 virus was also assessed. HI assays were performed according to the WHO manual for the laboratory diagnosis and virological surveillance of influenza with the use of guinea pig red blood cells under laboratory biosafety level 2 conditions [2]. Sera were pre-treated with receptor-destroying enzyme [RDE (II) “SEIKEN”; Denka Seiken Co. Ltd., Tokyo, Japan] and tested in two-fold serial dilutions starting at 1:10 to a final dilution of 1:1280. The HI endpoint titer was assessed as the reciprocal of the highest dilution of serum that completely inhibited hemagglutination. The WHO Collaborating Centre for Reference and Research on Influenza (WHO cc) in London provided reference virus strains and homologous antiserum, which were tested in each assay. An HI titer ≥40 was considered protective against the tested virus strains [2]. We also calculated geometric mean titer (GMT) at each observation moment. For this purpose, a value of five was assigned to all HI titers <10. The serological response to TIV, measured by the HI antibody titer, was evaluated using the European Agency for the Evaluation of Medical Products (EMA) criteria for serological data assessment: seroprotection rate, seroconversion rate (HI ≥ 40 if HI < 10 pre-vaccination or ≥4-fold HI if HI ≥ 10 pre-vaccination), and GMT fold change (GMT ratio of post-vaccination to pre-vaccination titer). At least one of the criteria had to be met to confirm the vaccine immunogenicity: seroprevalence rate > 70%, seroconversion rate > 40%, or average GMT fold increase > 2.5 [9].

### 2.3. Statistical Analysis

All statistical analyses were performed using R version 4.1.2 [10]. Descriptive statistics are reported for each baseline variable as the participant count and the corresponding percentage, in total and by vaccination history at baseline for each season. Seroprevalence (% HI titer ≥ 40) and the geometric mean of the antibody titer (GMT) with a 95% confidence interval (95% CI) were estimated at the three points of blood collection in each season for each assessed virus. The seroconversion of the antibody titer was determined 1 month and 6 months after vaccination considering EMA criteria. To measure the effect of vaccination history on immunogenic response to TIV, we estimated seroprevalence and GMT ratios between frequently and occasionally vaccinated individuals for each observation time and virus assessed using a generalized estimation equation model further adjusted for age, sex, and the presence of chronic disease.

## 3. Results

### 3.1. Study Population

A total of 74 and 72 TIV-vaccinated healthcare workers (HCWs) agreed to participate during the 2017/2018 and 2018/2019 influenza seasons, respectively. The mean age of the HCWs recruited during the 2017/2018 season was 50 years (range 34–69 years); 79.7% were female, and 14.9% had chronic diseases. During the 2018/2019 season, the mean age of the HCW was 52 years (range 33–70 years); 79.2% were female, and 26.8% had chronic diseases (Table 1). A total of 49 HCWs participated in both seasons (mean age 50, range 34–69 years, 73.5% female, 22.4% with chronic diseases) (see Appendix A).

During the 2017/2018 period, 32 (43.2%) of the participants were classified as frequently vaccinated (≥3 vaccines), and 19 (25.6%) had been vaccinated with the 2009 PV. During the 2018/2019 period, 43 (59.7%) of the participants were classified as frequently vaccinated (≥3 vaccines), and 23 (31.9%) had been vaccinated with the 2009 PV (Table 1).

### 3.2. Preexisting Antibodies Before Seasonal Vaccination

The presence of preexisting antibodies (HI titer ≥ 40) against influenza was identified for influenza A subtypes and both influenza B lineages. During the 2017/2018 season, the highest seroprevalence (41.9%; CI 95% 30.5–53.9) and GMT (25.3; CI 95% 20.3–31.5) were observed for A(H3N2), while the lowest seroprevalence and GMT were noted for influenza B/Yamagata (4.1%; CI 95% 0.8–11.4 and 8.4; CI 95% 7.3–9.8, respectively) (Table 2). In the 2018/2019 season, the preexisting seroprevalence and GMT were also higher for A(H3N2) (45.8%; CI 95% 34.0–58.0 and 26.4; CI 95% 20.8–33.6, respectively), and the lowest seroprevalence and GMT were observed for A(H1N1) and B/Victoria, respectively (Table 2). A more frequent presence of preexisting antibodies to influenza B and a higher GMT were observed in individuals who had been frequently vaccinated. Conversely, the presence of preexisting antibodies that provide protection against influenza A and a higher GMT were more commonly seen in those who had only been classified as occasionally vaccinated (Table 2).

### 3.3. Immunogenicity of the Seasonal Influenza Vaccine

The EMA’s criteria for vaccine assessment have been partially met for the influenza vaccine one month following vaccination. With regard to influenza A(H3N2), the seroprevalence rate one month following vaccination met the EMA criteria for the overall participants, reaching 70.3% (95% CI 58.5–80.3) and 81.9% (95% CI 71.1–90.0) during the 2017/18 and 2018/19 seasons, respectively. Although the seroconversion rate did not satisfy the criteria set by the EMA (>40%), a GMT fold increase greater than 2.5 was observed for A(H3N2) during the 2018/2019 season (Table 2). Among participants who had received occasional vaccinations, the seroprevalence for A(H3N2) one month after vaccination fulfilled the EMA criteria (76.2%; CI 95% 60.5–87.9 and 82.8%; CI 95% 64.2–94.2) in both seasons and in 2018/2019 for those who had received frequent vaccinations (81.4%; CI 95% 66.6–91.6) (Table 2). In the 2018/2019 season, one month after vaccination, seroprevalence reached 70% or above for A(H1N1), thus meeting the EMA criteria (Table 2).

The seroprevalence and GMT peaked at one month following vaccination, reaching higher levels for influenza A(H3N2) during both seasons (70.3%; CI 95% 58.5–80.3 and 81.9%; CI 95% 71.1–90.0, respectively). The seroconversion rate was higher for A(H1N1) during 2017/2018 (21.6% (CI 95% 12.9–32.7) and for A(H3N2) in 2018/2019 (36.1%; CI 95% 25.1; 48.3) (Table 2).

The occasionally vaccinated participants demonstrated a higher prevalence of protective antibodies and GMT for influenza A(H1N1)pdm09 and A(H3N2) throughout the study period.

Individuals who received frequent seasonal vaccinations demonstrated a higher seroprevalence rate, seroconversion, and GMT for influenza B Victoria at one month following vaccination during the 2018/2019 season. Despite the absence of B Yamagata in the TIV composition, a comparable pattern was observed to that of B Victoria during both seasons (Table 2 and Appendix A).

### 3.4. Immunity Extent Six Months After Seasonal Influenza Vaccination

The presence of protective antibodies against influenza was detectable six months following vaccination, despite the observation of a decline in seroprevalence and GMT over time. Often, the seroprevalence values were comparable to the preexisting antibody levels prior to vaccination. The seroprevalence and GMT for influenza A subtypes six months following vaccination were higher among those who had occasionally received the TIV. With regard to influenza B, both lineages exhibited higher seroprevalence and GMT in individuals who were frequently vaccinated compared to those who were occasionally vaccinated (Table 2).

An increase in the seroconversion rate was observed six months after vaccination for the influenza (sub)type in circulation during the follow-up period, which is likely attributable to virus exposure. This was particularly evident for influenza B Yamagata during the 2017/2018 season (Table 2).

### 3.5. 2009 Pandemic Vaccine and Immunogenic Response to A(H1N1) Subtype Viruses After Seasonal Influenza Vaccination

One month following vaccination, the seroprevalence of protective antibodies for TIV A(H1N1)pdm09 was similar between 2009PV recipients and non-recipients. However, the seroconversion rate was 40% higher in the non-recipient group during 2017/2018 (Table 3). Six months after vaccination, the discrepancy in seroprevalence and GMT regarding 2009PV uptake was less pronounced, although a degree of variability was observed across the two seasons included in the follow-up study (Table 3).

The seroprevalence, seroconversion, and GMT of protective antibodies against the influenza A(H1N1)pdm09 virus (A/California/7/2009) at one and six months following vaccination was comparable between participants, irrespective of 2009 PV vaccination (Table 3).

### 3.6. Antibody Dynamics During the Two Seasons

During the two seasons’ follow-up periods, 49 participants participated in both seasons. The serological evaluation for these participants comprised six observation points. A booster of the GMT and seroprevalence against the influenza vaccine viruses was observed one month after each seasonal vaccine administration for influenza A subtypes and influenza B both lineages. The seroprevalence and GMT for the 2017/2018 season were 49.0% (95% CI 34.4–63.7) and 34.7 (95% CI 25.7–46.9) and 69.4% (95% CI 54.6–81.7) and 48.1 (95% CI 37.0–62.5) for A(H1N1) and A(H3N2), respectively. The values for influenza B/Victoria seroprevalence and GMT were 53.1% (CI 95% 38.3–67.5) and 29.3 (CI 95% 23.5–36.5). Six months after vaccination, the seroprevalence and GMT decreased to values similar to those observed prior to vaccination: 14.6% (CI 95% 6.1–27.8) and 14.6 (CI 95% 12.0–17.6) and 52.1% (CI 95% 37.2–66.7) and 28.3 (CI 95% 22.8–35.1) for A(H1N1) and A(H3N2), respectively. With regard to influenza B/Victoria, the seroprevalence and GMT were 27.1% (CI 95% 15.3–41.8) and 19.7 (CI 95% 16.2–24.0). These values were sustained until 12 months after vaccination in 2017/2018. One month following the 2018/2019 annual TIV, the GMT and seroprevalence increased to values comparable to those observed in the previous season (Appendix A). Despite the absence of influenza B/Yamagata from the vaccine composition, an increase in seroprevalence and GMT was observed one month after vaccination. Exceptionally, the seroprevalence and GMT increased six months after the seasonal vaccination due to exposure to the virus during the 2017/2018 season (Appendix A).

For A(H1N1) and A(H3N2), occasional vaccination was associated with higher seroconversion, protective antibody seroprevalence, and GMT in most instances. During the 2017/2018 season, the ratio of antibody titers was 2.2 (95% CI 1.3–3.7) and 1.7 (95% CI 1.1–2.6) times higher for individuals who received occasional vaccinations, one month following the administration of the vaccine, for A(H1N1) and A(H3N2), respectively (Figure 1 and Appendix A). With regard to influenza B Victoria, it was observed that at the majority of seroevaluation points, the occasionally vaccinated group exhibited lower titers, with a notable decline in the second season (0.565; CI 95% 0.354–0.900) (Figure 1). A similar pattern was observed for influenza B Yamagata, where the occasionally vaccinated group exhibited consistently lower GMTs, with a notable decline during the 2018/2019 season one month after vaccination (Figure 1 and Appendix A).

## 4. Discussion

The study demonstrated that the seasonal influenza vaccine induced an immunological response against influenza A and B viruses, with a peak of seroprevalence and protective antibody titers one month after the vaccination in each season, irrespective of the previous seasonal vaccination frequency. During both influenza seasons included in the follow-up study, the annual TIV vaccination restored the protection antibody levels for influenza A subtypes and influenza B one month after vaccination. Other authors have previously demonstrated that antibody levels are boosted one month after receiving the seasonal influenza vaccination, regardless of the frequency of previous seasonal vaccinations [11,12,13].

Despite the fact that the TIV included only the influenza B Victoria virus, an increase in influenza B Yamagata seroprevalence and antibody titers was also detected. This demonstrates the heterotypic reactivity between both influenza B lineages. This phenomenon has been previously described in other studies [14], which demonstrated that the post-vaccination serological response elicits cross-reactive antibodies against both lineages, irrespective of the influenza B lineage vaccine strain employed. However, this response may not be sufficient to provide an adequate level of serological protection in the event of a vaccine mismatch [14,15]. The present study demonstrated that TIV induced the production of protective antibodies against the influenza B Yamagata lineage, although at a lower seroprevalence and antibody titer than for the B Victoria lineage. Influenza B Yamagata has not been circulating since 2020, so the importance of cross-reactive antibodies is unclear at the moment.

A significant finding of our study was the effect of the TIV on lower antibody seroprevalence and booster titers one month after vaccination, in frequently vaccinated individuals compared to those occasionally vaccinated. Frequently vaccinated individuals exhibited a reduced response to the TIV one month after vaccination when compared to those who had received occasional vaccinations (≤2 TIV doses since 2015/2016). This was evidenced by a reduction in seroprevalence and antibody titers, particularly for influenza A subtypes.

In the 1970s, T. Hoskins published the first description of the repeated vaccination effect on influenza outbreaks in schoolchildren in the United Kingdom. This demonstrated that vaccination in a previous season could indirectly affect the risk of infection in a future season. Conceivably, prior vaccinations can block opportunities to acquire a broader and more durable influenza protection effect after infection (i.e., the infection block effect) [16].

Some authors have demonstrated that the immunogenicity of the vaccine is enhanced by recent virus infection but attenuated by recent vaccination [17]. Although vaccination in the previous year attenuates the vaccine immune response, vaccination in two consecutive years provides superior protection to that afforded by no vaccination whatsoever [12,18].

A reduced response to influenza A subtypes in individuals who have received frequent vaccinations has been hypothesized previously by other authors. This phenomenon can be explained by two theories: the “negative interference” hypothesis and the antigenic distance hypothesis (ADH). These theories suggest that repeated vaccination with similar influenza strains may lead to antigen depletion by preexisting antibodies and a subsequent reduction in the immune response to vaccination [19]. During both seasons, old and new influenza A(H1N1) and A(H3N2) vaccine strains were antigenically and genetically similar, whereas the old and new influenza B Victoria strains displayed differences [20,21].

Nevertheless, other researchers have proposed that preexisting antibodies enhance the immune response to vaccination [13], partly due to the higher pre-vaccination antibody levels observed in some studies [22]. Repeat influenza vaccination can result in higher baseline titers and lower-magnitude fold changes, particularly when the vaccine strains are similar in consecutive seasons. Preexisting antibody titers are correlated with greater odds of achieving a seroprotective titer, although they are inversely correlated with seroconversion [13,23].

Our study is one of the few studies that allowed for the follow-up of HCWs over two consecutive influenza seasons to assess acquired immunity achieved following the administration of the TIV for both influenza A subtypes and both B lineages. Other studies that evaluated the impact of previous vaccination on HCWs demonstrated an association between vaccine serologic response and prior vaccination. Consistent with other studies examining HCW responses to influenza vaccines, we observed that the serological response to the influenza vaccine was inversely associated with the number of prior vaccinations for influenza A subtypes [22,24,25]. However, for influenza B, our results strongly suggest the need for annual TIV to boost and maintain seroprevalence and higher antibody titers against both influenza B lineages.

The findings of our study indicate that antibodies against influenza were maintained in both frequently and occasionally vaccinated individuals throughout the winter season and were still detected 6 and 12 months after vaccination. Despite this, a decline was noted in seroprevalence and antibody titer six months following vaccination. It is noteworthy that an increase in antibody titers was observed over the course of the season for the influenza strains that were circulating at the time, as was the case with influenza B Yamagata six months after the vaccination follow-up took place during the 2017/2018 season. This increase is likely to have been due to exposure to the epidemic virus.

During the last influenza pandemic, a monovalent vaccine was available for HCWs during the 2009/2010 season. The data demonstrated that the vaccination of HCWs with the 2009 pandemic vaccine did not significantly influence the antibody response several years later to the A(H1N1)pdm09 component of the trivalent influenza vaccine (TIV). However, a higher seroconversion rate for the seasonal A(H1N1)pdm09 vaccine strain was observed one month after vaccination among participants who had not received the 2009PV. Other authors have indicated that there is no evidence of significant differences in antibody responses and GMT against A(H1N1)pdm09 in HCWs who received the 2009 PV and subsequently received the TIV and those who received the TIV on an occasional basis [11].

The main limitation of our study was the small sample size. However, we included active adult HCWs from all age spans. The generalization of the findings of the study to other population groups should be made with caution. The principal strength of the study was the opportunity to conduct a two-season follow-up (24 months) of the same individuals who had received multiple vaccinations, with the aim of assessing immunity at six time points over the entire period. The study has valuable data because it was developed in the two previous seasons right before the introduction of the COVID-19 vaccine and the massive co-administration of both vaccines against influenza and COVID-19 in HCWs since 2020. The present study was the only longitudinal study carried out in HCWs in Portugal.

Further research is required with a larger number of participants and over multiple years in order to gain a deeper understanding of immunogenicity and influenza vaccine effectiveness, as well as the impact of prior vaccination. Given the existence of overlapping risk groups and the temporal proximity of annual vaccine campaigns for influenza and SARS-CoV-2, it is imperative to assess the potential for immune interference with regard to the effectiveness of these vaccines.

## 5. Conclusions

The results of our study indicate that the serological response following the vaccination of HCWs is influenced by their previous vaccination history and exhibits distinct patterns for each influenza type. A history of repeated seasonal vaccination was associated with a reduction in the immunological response against both the influenza A(H3N2) and A(H1N1)pdm09 viruses. In contrast, repeated vaccination with the influenza B vaccine resulted in a higher seroprevalence and antibody titer.

It has been demonstrated that previous vaccinations can influence the immune response. Nevertheless, the administration of the influenza vaccine on an annual basis has been shown to effectively enhance immunity. Irrespective of the frequency of influenza vaccination, an increase in seroprevalence and antibody titers was observed following vaccination for all virus components. Antibody titers wane over time but are detectable up to 12 months after vaccination.

It is imperative that annual influenza vaccination be undertaken in order to re-establish and maintain antibody titers against influenza A and B during the season, even in the absence of a change in the vaccine strains’ composition. The current recommendations for influenza vaccination are for annual vaccination, with no evidence to suggest that repeated seasonal vaccination impairs immunity [26].

Serological studies are an invaluable source of information, providing crucial insights that inform recommendations for influenza vaccine composition and contribute to the management of vaccination campaigns.

## Figures and Tables

**Figure 1 vaccines-12-01218-f001:**
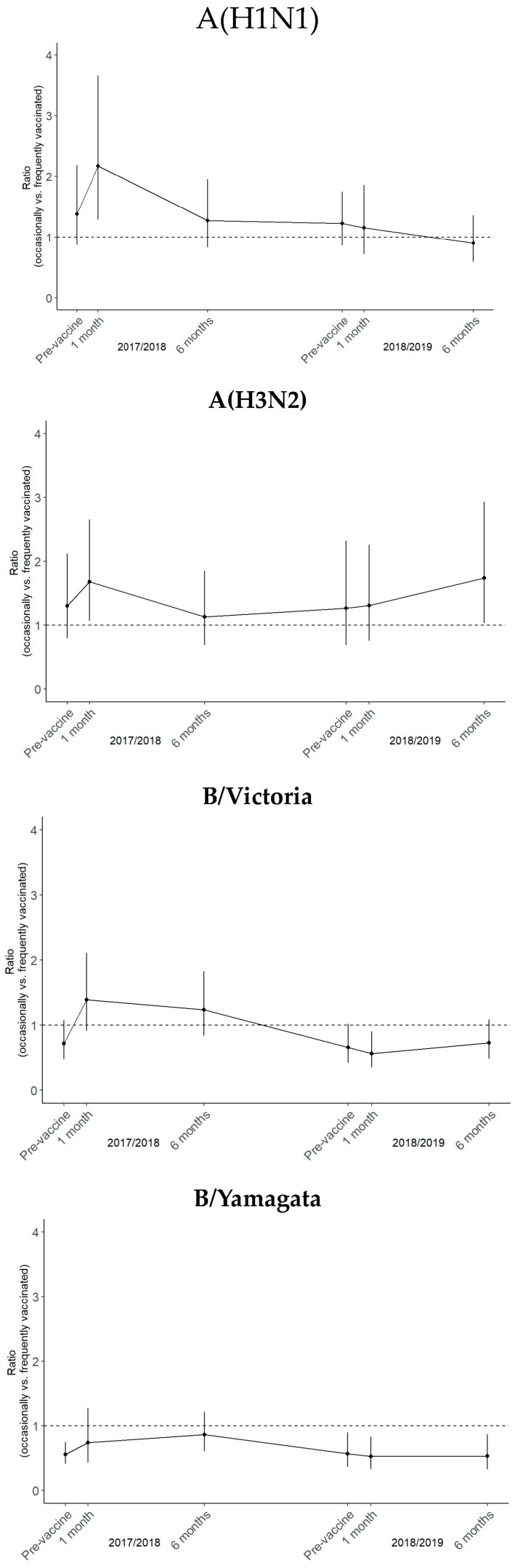
GMT ratios (occasionally vs. frequently vaccinated HCWs) with confidence intervals (CI 95%) (vertical bars) for influenza A(H1N1), A(H3N2), B Victoria, and B Yamagata. GMT ratios were adjusted for age and gender, and the 2017/2018 pre-vaccination GMT was considered for reference. For the 49 participants that repeatedly received the seasonal TIV during 2017/2018 and 2018/2019 seasons, 6 time points were considered during the follow-up period (pre-vaccine 2027/2028, 1 and 6 months post-vaccination, pre-vaccine 2018/2019, and 1 and 6 months post-vaccination).

**Table 1 vaccines-12-01218-t001:** Demographic characteristics and vaccination history of the participants included in the study during 2017/2018 and 2018/2019 seasons.

			Vaccination *n* (%)		
		Total *n* (%)	Frequent (≥3 vaccines)	Occasional (≤2 vaccines)	2009 Pandemic Vaccine
2017/2018	N	74	32	42	19
	Age group (yr) *n* (%)				
	20–49	34 (45.9%)	14 (43.8%)	20 (47.6%)	11 (57.9%)
	50–69	40 (54.1%)	18 (56.2%)	22 (52.4%)	8 (42.1%)
	Mean (SD)	50.3 (8.6)	50.1 (9.2)	50.5 (8.2)	49.053 (7.8)
	Range	34–69	34–68	35–69	34–61
	Sex *n* (%)				
	Female	59 (79.7%)	23 (71.9%)	36 (85.7%)	14 (73.7%)
	Male	15 (20.3%)	9 (28.1%)	6 (14.3%)	5 (26.3%)
	Chronic disease *n* (%)				
	No	63 (85.1%)	28 (87.5%)	35 (83.3%)	16 (84.2%)
	Yes	11 (14.9%)	4 (12.5%)	7 (16.7%)	3 (15.8%)
	ILI *n* (%)				
	Influenza positive	1 (7.1%)	0 (0.0%)	1 (9.1%)	1 (50.0%)
	Negative	13 (92.9%)	3 (100.0%)	10 (90.9%)	1 (50.0%)
2018/2019	N	72	43	29	23
	Age group (yr) *n* (%)				
	20–49	30 (41.7%)	18 (41.9%)	12 (41.4%)	11 (47.8%)
	50–70	42 (58.3%)	25 (58.1%)	17 (58.6%)	12 (52.2%)
	Mean (SD)	52.2 (8.5)	52.4 (8.9)	51.8 (8.0)	51.7 (7.9)
	Range	33–70	35–70	33–66	35–63
	Sex *n* (%)				
	Female	57 (79.2%)	32 (74.4%)	25 (86.2%)	17 (73.9%)
	Male	15 (20.8%)	11 (25.6%)	4 (13.8%)	6 (26.1%)
	Chronic disease *n* (%) *				
	No	52 (73.2%)	32 (74.4%)	20 (71.4%)	16 (72.7%)
	Yes	19 (26.8%)	11 (25.6%)	8 (28.6%)	6 (27.3%)
	ILI *n* (%)				
	Influenza positive	2 (15.4%)	1 (14.3%)	1 (16.7%)	1 (14.3%)
	Negative	11 (84.6%)	6 (85.7%)	5 (83.3%)	6 (85.7%)

* Three participants lacked data on chronic diseases. During the 2017/2018, 1 case was excluded from the last sampling point (6 months) after testing positive for influenza. During the 2018/2019 season, 2 participants tested positive for influenza and were also excluded from the last sampling point (6 months).

**Table 2 vaccines-12-01218-t002:** Immunological response to the seasonal trivalent influenza vaccine during 2017/2018 and 2018/2019 for frequently and occasionally vaccinated individuals. Antibodies against influenza AH1N1, AH3N2, BVictoria, and B Yamagata were measured by hemagglutination inhibition assay before vaccination and 1 month and 6 months after vaccination in each season.

		Influenza Viruses										
	AH1N1				AH3N2				B Victoria			
	Total	Frequent vacc.	Occasional vacc.	Ratio	Total	Frequent vacc.	Occasional vacc.	Ratio	Total	Frequent vacc.	Occasional vacc.	Ratio
2017/2018 (N)	74	32	42			32	42			32	42	
Seroprotection rate % (95% CI)												
Pre-vaccination	23.0 (14;34.2)	15.6 (5.3;32.8)	28.6 (15.7; 44.6)	1.8 (0.7; 4.7)	41.9 (30.5; 53.9)	43.8 (26.4;62.3)	40.5 (25.6;56.7)	0.9 (0.5; 1.6)	32.4 (22;44.3)	43.8 (26.4; 62.3)	23.8 (12.1;39.5)	0.5 (0.3; 1.1)
1 month post-vaccination	55.4 (43.4;67)	40.6 (23.7;59.4)	66.7 (50.5;80.4)	1.6 (1.0;2.6)	70.3 (58.5;80.3)	62.5 (43.7;78.9)	76.2 (60.5;87.9)	1.2 (0.9; 1.7)	52.7 (40.7; 64.4)	50.0 (31.9; 68.1)	54.8 (38.7;70.2)	1.1 (0.7; 1.7)
6 months post-vaccination	12.3 (5.8;22.1)	6.2 (0.8;20.8)	17.1 (7.2;32.1)	2.7 (0.6;12.3)	45.9 (34.3;57.9)	40.6 (23.7;59.4)	50.0 (34.2;65.8)	1.2 (0.7;2)	27.4 (17.6;39.1)	25.0(11.5;43.4)	29.3 (16.1;45.5)	1.2 (0.5; 2.5)
Seroconversion rate % (95% CI)												
1 month post-vaccination	21.6 (12.9;32.7)	9.4 (2;25)	31.0 (17.6;47.1)	3.3 (1;10.6)	18.9 (10.7;29.7)	9.4 (2;25)	26.2 (13.9;42)	2.8 (0.8;9.2)	9.5 (3.9;18.5)	0.0 (0;10.9)	16.7 (7;31.4)	*
6 months post-vaccination	2.7 (0.3;9.4)	3.1 (0.1;16.2)	2.4 (0.1;12.6)	0.8 (0;11.7)	12.2 (5.7;21.8)	15.6 (5.3;32.8)	9.5 (2.7;22.6)	0.6 (0.2;2.1)	9.5 (3.9;18.5)	0.0 (0;10.9)	16.7 (7;31.4)	*
GMT (95% CI)												
Pre-vaccination	20.0 (16.7;24)	16.5 (12.6;21.4)	23.2 (18.1;29.8)	1.4 (0.7;2.7)	25.3 (20.3;31.5)	22.8 (16.1;32.1)	27.4 (20.3;36.9)	1.2 (0.7;2.1)	20.4 (17.2;24.1)	24.8 (18.6;33.1)	17.5 (14.4;21.4)	0.7 (0.4;1.3)
1 month post-vaccination	36.4 (29.1;45.7)	23.3 (17.6;30.8)	51.2 (37.8;69.5)	2.2 (1.3;3.6)	48.7 (38.8;61.1)	35.9 (26.1;49.3)	61.4 (45.1;83.7)	1.7 (1.1;2.6)	28.8 (24.5;34)	27.1 (20.6;35.7)	30.2 (24.5;37.2)	1.1 (0.7;1.9)
6 months post-vaccination	14.3 (12.4;16.5)	13.0 (10.5;16)	15.5 (12.7;19)	1.2 (0.6;2.5)	25.8 (21.6;30.9)	23.8 (18;31.4)	27.6 (21.6;35.2)	1.2 (0.7;2.0)	21.0 (17.8;24.7)	21.3 (16.1;28.2)	20.7 (16.8;25.4)	1.0 (0.5;1.8)
GMT ratio (95% CI)												
1 month post-vaccination	1.8 (1.5;2.2)	1.4 (1.2;1.7)	2.2 (1.6;3.0)		1.9 (1.6;2.3)	1.6 (1.3;1.9)	2.2 (1.8;2.8)		1.4 (1.2;1.7)	1.1 (1.0;1,2)	1.7 (1.3;2,2)	
6 months post-vaccination	0.7 (0.6–0.9)	0.8 (0.6;1.0)	0.7 (0.5;0.9)		1.0 (0.8;1.3)	1.0 (0.8;1.4)	1.0 (0.7;1.3)		1.0 (0.9;1,2)	0.9 (0.7;1.0)	1.2 (0.9;1.6)	
2018/2019 (N)	70	41	29		70	41	29		70	41	29	
Seroprotection rate % (95% CI)												
Pre-vaccination	20.8 (12.2;32)	16.3 (6.8;30.7)	27.6 (12.7;47.2)	1.7 (0.7;4.2)	45.8 (34.0;58.0)	39.5(25;55.6)	55.2(35.7;73.6)	1.4 (0.9;2.3)	23.9 (14.6;35.5)	25.6 (13.5;41.2)	21.4(8.3;41)	0.8 (0.3;2.0)
1 month post-vaccination	73.6 (61.9;83.3)	79.1(64.0;90.0)	65.5(45.7;82.1)	0.8 (0.6;1.1)	81.9 (71.1;90)	81.4 (66.6;91.6)	82.8 (64.2;94.2)	1.0 (0.8;1.3)	56.3 (44;68.1)	62.8 (46.7;77)	46.4 (27.5;66.1)	0.7 (0.5;1.2)
6 months post-vaccination	27.1 (17.2;39.1)	26.2 (13.9;42)	28.6 (13.2;48.7)	1.1 (0.5;2.4)	64.3 (51.9;75.4)	59.5 (43.3;74.4)	71.4 (51.3;86.8)	1.2 (0.9;1.7)	46.4 (34.3;58.8)	54.8 (38.7;70.2)	33.3 (16.5;54)	0.6 (0.3;1.1)
Seroconversion rate % (95% CI)												
1 month post-vaccination	31.9 (21.4;44)	34.9 (21;50.9)	27.6 (12.7;47.2)	0.8 (0.4;1.6)	36.1 (25.1;48.3)	34.9(21.0;50.9)	37.9(20.7;57.7)	1.1 (0.6;2)	23.9 (14.6;35.5)	27.9 (15.3;43.7)	17.9 (6.1;36.9)	0.6 (0.2;1.6)
6 months post-vaccination	2.8 (0.3;9.7)	2.3 (0.1;12.3)	3.4 (0.1;17.8)	1.5 (0.1;22.8)	22.2 (13.3;33.6)	23.3 (11.8;38.6)	20.7 (8;39.7)	0.9 (0.4;2.2)	13.9 (6.9;24.1)	18.6 (8.4;33.4)	6.9 (0.8;22.8)	0.4 (0.1;1.6)
GMT (95% CI)												
Pre-vaccination	17.5 (15;20.3)	17.6 (14.6;21.2)	17.3 (13.3;22.6)	1.0 (0.5;1.9)	26.4 (20.8;33.6)	22.8 (17.1;30.3)	33.0(21.5;50.7)	1.5 (0.9;2.5)	15.8 (13;19.2)	16.8 (13.3;21.1)	14.5 (10.2;20.7)	0.9 (0.4;1.7)
1 month post-vaccination	40 (33.4;48)	40.7 (32.4;50.9)	39.1 (28.4;53.7)	1.0 (0.6;1.5)	75.5 (58.8;97)	68.1 (49.5;93.6)	88.0 (57.8;134)	1.3 (0.9;1.8)	33.9 (27.6;41.7)	38.1 (29.5;49.3)	28.3 (19.9;40.3)	0.7 (0.5;1.2)
6 months post-vaccination	18.1 (15.5;21.2)	19.4 (16.1;23.3)	16.4 (12.3;21.9)	0.8 (0.4;1.6)	46.7 (36.5;59.7)	41.3 (29.3;58.3)	56 (39.5;79.3)	1.4 (0.9;2.0)	24.5 (20.4;29.4)	26.5 (20.8;33.8)	21.6 (16.2;28.8)	0.8 (0.5;1.4)
GMT ratio (95% CI)												
1 month post-vaccination	2.3 (2.0;2.7)	2.3 (1.9–2.8)	2.3 (1.7;2.9)		2.9 (2.3;3.5)	3.0 (2.3;3.9)	2.7 (2.0;3.6)		2.1 (1.8;2.6)	2.3 (1.8;2.9)	2.0 (1.5;2.6)	
6 months post-vaccination	1.0 (0.9;1.2)	1.1 (0.9;1.3)	1.0 (0.8;1.2)		1.8 (1.5;2.1)	1.8 (1.4;2.3)	1.7 (1.3;2.1)		1.6 (1.3;1.8)	1.6 (1.3;2.0)	1.4 (1.1;2.9)	

* Not possible to calculate (denominater was zero).

**Table 3 vaccines-12-01218-t003:** Immunological response to influenza A/Michigan/45/2015 (A H1N1pdm09 2017/18 and 2028/19 vaccine virus) and A/California/7/2009 (AH1N1pdm09 2009PV virus) after seasonal trivalent influenza vaccination during the 2017/2018 and 2018/2019 seasons.

		Influenza A(H1N1)					
		AH1N1pdm09 (2017/18 and 2028/19 Vaccine Virus) (A/Michigan/45/2015)			AH1N1pdm09 (2009PV) (A/California/7/2009)		
		2009PV vaccine recipients	2009PV vaccine non-recipients	Ratio (non-recipients/recipients)	2009PV vaccine recipients	2009PV vaccine non-recipients	Ratio (non-recipients/recipients)
2017/2018 N = 54	N	19	35		19	35	
	Seroprotection rate *n* (%; 95% CI)						
	Pre-vaccination	26.3 (9.1;51.2)	20.0 (8.4;36.9)	0.8 (0.3;2.1)	57.9 (33.5;79.7)	71.4 (53.7;85.4)	1.2 (0.8;1.9)
	1 month post-vaccination	52.6 (28.9;75.6)	51.4 (34;68.6)	1.0 (0.6;1.7)	100.0 (81.5;100)	91.4 (76.9;98.2)	0.9 (0.8;1.0)
	6 months post-vaccination	5.6(0.1;27.3)	17.1 (6.6;33.6)	3.1 (0.4;23.7)	31.6 (12.6;56.6)	25.7 (12.5;43.3)	0.8 (0.3;1.8)
	Seroconversion rate *n* (%; 95% CI)						
	1 month post-vaccination	15.8 (3.4;39.6)	22.9 (10.4;40.1)	1.4 (0.4;4.8)	33.3 (13.3;59.0)	20 (8.4;36.9)	0.6 (0.2;1.6)
	6 months post-vaccination	5.3 (0.1;26)	2.9 (0.1;14.9)	0.5 (0;8.2)	0.0 (0.0;17.6)	0.0 (0.0;10.0)	0.0
	GMT (95% CI)						
	Pre-vaccination	20.0 (13;30.7)	18.8 (14.1;25.2)	0.9 (0.5;1.8)	37.2 (25.0;55.4)	39.2 (30.6;50.3)	1.1 (0.7;1.7)
	1 month post-vaccination	32.1 (20.3;50.8)	36.2 (25.1;52.2)	1.1 (0.7;1.8)	80.0 (56.2;113.9)	71.0 (53.1;95.1)	0.9 (0.6;1.2)
	6 months post-vaccination	14.9 (11.6;19.3)	14.6 (11.5;18.5)	1.0 (0.5;2.1)	22.3 (16.9;29.5)	20.0 (15.9;25.2)	0.9 (0.5;1.6)
	GMT ratio (95% CI)						
	1 month post-vaccination	1.6	1.9		2.2	1.8	
	6 months post-vaccination	0.7	0.8		0.6	0.5	
2018/2019 N = 70	N	23	47		23	47	
	Seroprotection rate *n* (%; 95% CI)						
	Pre-vaccination	17.4 (5.0;38.8)	21.3 (10.7;35.7)	1.2 (0.4;3.5)	43.5 (23.2;65.5)	42.6 (28.3;57.8)	1.0 (0.6;1.7)
	1 month post-vaccination	87.0 (66.4;97.2)	66.0 (50.7;79.1)	0.8 (0.6;1)	69.6 (47.1;86.8)	70.2 (55.1;82.7)	1.0 (0.7;1.4)
	6 months post-vaccination	31.8 (13.9;54.9)	26.1 (14.3;41.1)	0.8 (0.4;1.8)	31.8 (13.9;54.9)	47.8 (32.9;63.1)	1.5 (0.8;3)
	Seroconversion rate *n* (%; 95% CI)						
	1 month post-vaccination	34.8 (16.4;57.3)	29.8 (17.3;44.9)	0.9 (0.4;1.7)	13 (2.8;33.6)	21.3 (10.7;35.7)	1.6 (0.5;5.4)
	6 months post-vaccination	4.3 (0.1;21.9)	2.1 (0.1;11.3)	0.5 (0;7.5)	0.0 (0.0;14.8)	8.5 (2.4;20.4)	*
	GMT (95% CI)						
	Pre-vaccination	18.8 (15.1;23.4)	16.8 (13.6;20.6)	0.9 (0.5;1.7)	26.2 (18.6;37.1)	23.9 (18.7;30.5)	0.9 (0.5;1.6)
	1 month post-vaccination	46.5 (35.5;60.9)	37.2 (29;47.6)	0.8 (0.5;1.2)	46.5 (31.4;68.9)	47.7 (37.4;61.0)	1.0 (0.7;1.5)
	6 months post-vaccination	21.9 (17.2;27.9)	17.0 (13.9;20.9)	0.8 (0.4;1.5)	24.7 (17.5;34.8)	29.8 (23.5;37.8)	1.2 (0.7;2.1)
	GMT ratio (95% CI)						
	1 month post-vaccination	2.5	2.2		1.8	2.0	
	6 months post-vaccination	1.2	1.0		0.9	1.2	

* Not possible to calculate (denominater was zero).

## Data Availability

The data presented in this study are available on request from the corresponding author. The data are not publicly available due to confidential procedures stated in the Informed Consent.

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
