# Peer review of "Immune Response Against Influenza in a Cohort of Repeatedly Vaccinated Adults During the 2017/2018 and 2018/2019 Seasons"

_vaccines, 2024, doi:10.3390/vaccines12111218_

Round 1
Reviewer 1 Report
Comments and Suggestions for Authors
This clinical study aimed to assess the impact of frequent influenza vaccination on the immune responses to seasonal trivalent influenza vaccine in health care workers. This study is important for possible corrections of recommendations for vaccination of HCWs, and owing to the known phenomenon of reducing vaccine effectiveness after repeated vaccinations with inactivated seasonal influenza vaccines. The study presents some new findings that may be interesting for public health authorities, but there are some issues that need to be addressed prior publication.
Major comment:
1. The major issue of this study is that the two subgroups were split based on the empirical sense what “occasional” and “frequent” vaccination mean. The timeframe for determining vaccination status is from 2015/2016 to 2018/2019, e.g. 15/16, 16/17, 17/18, 18/19 – four seasons, and two of them are within the study period. And in one group of occasionally vaccinated individuals could be recruited those who never received vaccine, or received them two times in the two previous seasons. These subjects are supposed to respond very differently to the TIV vaccination, but they were combined within one group, obviously resulting in significant variation of immunogenicity among participants from this group. What is the rationale to treat HCWs with two prior vaccinations in previous 3-4 seasons as occasional, not frequent?
Minor comments:
1. Since some participants were recruited in both seasons, it is likely that some may have been transitioned from one cohort to the other, but it is not clear from the Tables if this indeed occurred.
2. Please specify n the Methods the TIV used in this study. Was it split or subunit, adjuvanted or not?
3. Please correct “ocasional” in the tables.
4. The left column in Table II is incomplete.
5. It looks that the ILI-positive cases were included into immunogenicity analyses (Table II, number of participants is the same as in Table I). If infections occurred within the study period prior to blood sample collection, these subjects should be excluded from the immunogenicity analysis.
6. The immunogenicity results presented in the table would be easier to understand if they were presented graphically.
7. Figure 1 is difficult to read. It is not clear whether the variations in the GMT ratios are significant. And where do the deviations of the ratios come from?
8. Lane 19. “works” – please correct to “workers”
9. Lanes 137-138: word “respectively” should come after mentioning influenza seasons.
10. Supplementary figures are also very difficult to read.
Author Response
Please see the attachement with the response point-by-point.

Reviewer 2 Report
Comments and Suggestions for Authors
Review Guiomar et al.
Guiomar et al describe anti-influenza immune responses in health care workers who received the trivalent influenza vaccine in the seasons 2017/18 and 2018/19. They explore differences in responses between individuals who received the vaccine occasionally in the previous year and those that received it frequently. The findings seem to be in line with a number of publications that have shown that vaccination can dampen immune response to a certain extent. Importantly, the authors conclude that annual vaccination is important, especially to maintain seroprotective antibody levels against influenza B.
Main concern. Many concluding statements are made on differences in immune responses against the different influenza strains between occasionally and frequently vaccinated participants. In most cases, if not all, the 95% confidence intervals are largely overlapping and no p-values are given, Therefore, it is not clear which differences are statistically significant. In line 288 a statistically significant difference is indicated but it is not exactly clear to which numbers in Table 2 this refers to. The authors should add p-values to the tables. For differences that are not statistically significant different, the authors should reword the text to make clear that some of the findings could indicate trends but larger studies will be needed to demonstrate if the observed differences are true.
Lines 142-143&146-147. If 49 HCW participated in both seasons, were they counted twice? In other words, did the study actually enroll 146-49=97 HCW?
Line 272. Change 'was composed of' to 'contained' or 'included'.
Line 281. The authors could add that influenza B Yamagata has not been circulating since 2020, so the importance of cross-reactive antibodies is unclear at the moment.
There are five paragraphs on the possible explanations for diminished responses against influenza A in frequently vaccinated individuals. This can be significantly reduced in length.
Lines 343-345. On which numbers is this statement based? The effect was not seen in 2018/2019. Can the authors explain the differences that they observed between 2017/18 and 2018/19?
Do the authors have an explanation for the high pre-vaccination titers for the 2009 virus?
Comments on the Quality of English LanguageLines 46-49. It would be better to start the sentence with 'In Portugal, the vaccine is recommended etc.'
Lines 72-76 would be better placed at the beginning of the Results section than under Study design.
Line 147. Suggestion to add 'were collected' after 'serum samples'.
Table II does not fit the page. As a consequence, the left hand column is only partially readable. Note that the 4th section of 2017/2018 and 2018/2019 mentions '…..GMT ratio (95%)' but fold changes without 95%CI are indicated.
Line 187. 'The participants who were occasionally vaccinated,…' or 'The occasionally vaccinated participants, ….'
Line 193. Should his be 'a comparable pattern was observed for B Yamagata during both seasons'?
The title of Table III mentions 'in individuals that received the monovalent AH1N1pandemic 2009 vaccine', however data from both recipients and non-recipients of 2009PV are included in the table. Suggestion to delete 'in individuals that received the monovalent AH1N1pandemic 2009 vaccine' from the title.
Line 224, title Fig.1 and Title SS4. Replace 'uptake' to 'received'. In many other places where 'the (vaccine) uptake' is used, words like 'vaccination' or 'receiving/reception' should be preferred. Uptake is most often used in the sense of which percentage of the population is vaccinated. Please, check complete document.
Line 322. 'The study described here….' or 'Our study…..' instead of 'The study developed…'
Lines 369-370. It is not clear what the authors are trying to say. The first part of the sentence seems to be in contrast to the second part of the sentence.
Lines 372-374. Suggestion to say: 'Antibody titers wane over time, but are detectable up to 12 months after vaccination.'
Please, check document for double spaces.
Reference 2. WHO instead of Who, check abbreviation 'World Heal Organ'. This manual has 153 pages, which is different from p. 153.
Reference 3. The link seems to link to a website to report adverse events, not directly to the indicated publication.
Reference 5. Name of Journal is missing.
Reference 12, 13 and 23. Page numbers are missing.
Reference 21. It is not clear to which journal 'vol. 93, no. 12, pp. 133-141, 2018' refers to. Similar details are missing in reference 20.
Author Response
Please see the attachement with response point-by-point.

Reviewer 3 Report
Comments and Suggestions for Authors
In the manuscript entitled “Immune response against influenza in a cohort of repeatedly 2 vaccinated adults during 2017/2018 and 2018/2019 seasons”, the authors mainly investigated the impact of repeated influenza vaccine uptake on vaccine-induced immunity against influenza. There are major and minor concerns regarding this manuscript.
1) Only 146 participants were included for this survey. Please state how to calculate the minimum sample size of participants for this study. Please also state the inclusion and exclusion criteria for these participants in this survey.
2) Only data of 2017/2018 and 2018/2019 seasons, I think this is outdated. The updated data of recent years, especially during COVID-19 and after COVID-19 pandemic, should be more valuable.
3) Many similar studies have been published previously, and the innovation of this study is not sufficient. Authors should compare their work and provide the discrepancy.
4) All Tables should be formatted as three-line tables.
5) Figure 1, lack of legend.
Comments on the Quality of English LanguageMinor editing of English language required.
Author Response

(The authors gave the same response as above.)

Reviewer 4 Report
Comments and Suggestions for Authors
This appears to be a well constructed study, examining antibody responses to TIV vaccine, comparing persons who had <=2 recent flu vaccines to those who had >=3 doses. You also look at antibody response to the H1N1 component comparing persons who did and did not receive H1N1 monovalent flu immunization in 2009. Your hypothesis is that frequent prior flu immunization would blunt subsequent antibody response, as observed in a study you reference.
The differences you see are small, only significantly significant for titers one month post TIV. It is possible that your groups are not diverse enough to see a bigger difference, for example someone who had 2 prior flu immunizations vs 3 prior might not be expected to have different responses. Nevertheless, this information can be added to that from other studies (which you have appropriately referenced) and support a meta-analysis in the future.
It might be helpful to have bar charts to compare the titers of the different groups, the tables are lengthy and difficult to review. Perhaps those can be in a supplement or appendix.
I did not see in methods how you ascertained prior history of flu immunization. If it was by medical record or employee record, that would be terrific. If it was by self report, that should be noted in limitations.
When you were able to diagnose a case of flu infection in one of your subjects, did you control for that in subsequent analysis or did you omit that person from all subsequent analysis?
Some of the text was not clear to me. Line 35: "had been frequently vaccinated during the 2017/18 and 2018/19 seasons". I would word this "were categorized with a history of prior frequent vaccination" so it is clear those immunizations were not received in the years they were enrolled in your study.
Line 73: Please spell out INSA.
Line 88: What is a "followup moment"? Do you mean, they were excluded from the next time they would have had serology collected? As asked earlier, were they only excluded from that one or from the rest of the study?
Lines 144-146: Please word more clearly: e.g. ... were "classified as" frequently vaccinated... and "reported having" had the 2009...
Line 163: ... those 'classified as" occasionally vaccinated..
Line 174: unclear phrase "Despite the seroconversion rate did not meet"...
Lines 211-212: is this header 3.5?
Line 224: 49 participants "were classified as frequent recipients""
Line 250: again, the term "moment' is unclear
Your discussion is interesting, you point out what you are showing that is different from others. I would include as a limitation how you obtained flu vacc histories and that you dont have flu illness data.
Comments on the Quality of English Language
Some phrases are not clear, as noted above.
Author Response
Please see the attachment with the response point-by-point.

Round 2
Reviewer 1 Report
Comments and Suggestions for Authors
The authors adequately addressed all the issues raised during original peer review. I have no further comments
Reviewer 2 Report
Comments and Suggestions for Authors
Thank you for addressing my questions and comments. Although I am understand the explanation to indicate relative differences, I don't think that excludes that some of the findings are chance findings.
Reviewer 3 Report
Comments and Suggestions for Authors
The authors failed to fully address my question. For example, how to calculate the minimum sample size of participants for this study?Authors also mentioned "the study has valuable data because was developed right in the two previous seasons before the COVID-19 pandemic and right before the introduction of the COVID-19 vaccine" in the revised Discussion section.
